# Effectiveness of Heterologous and Homologous Ad26.COV2.S Vaccine Boosting in Preventing COVID-19-Related Outcomes Among Individuals with a Completed Primary Vaccination Series in the United States

**DOI:** 10.3390/vaccines13020166

**Published:** 2025-02-09

**Authors:** Mawuli Nyaku, Lara S. Yoon, Deborah Ricci, Lexie Rubens, Paige Sheridan, Monica Iyer, Thomas Zhen, Raymond A. Harvey, Ann Madsen

**Affiliations:** 1Johnson & Johnson Innovative Medicine, Data Science and Digital Health, Spring House, PA 19002, USA; dricci@its.jnj.com (D.R.); rharvey2@its.jnj.com (R.A.H.); 2Aetion Inc., New York, NY 10001, USAann.madsen@aetion.com (A.M.)

**Keywords:** COVID-19, adenovirus vaccine, booster vaccine, real-world data, real-world evidence

## Abstract

**Background/Objectives:** COVID-19 vaccines have significantly reduced the mortality and morbidity associated with SARS-CoV-2. In the fall of 2021, the U.S. Food and Drug Administration amended its emergency use authorization guidelines for COVID-19 vaccines to allow the administration of booster vaccine doses based on sound scientific evidence of the increase in effectiveness conferred by boosters. The effectiveness of the Ad26.COV2.S COVID-19 booster vaccine during the periods of Delta and Omicron variant dominance is unknown. This study used real-world data to estimate the effectiveness of booster heterologous or homologous Ad26.COV2.S vaccination compared to that of a primary Ad26.COV2.S or mRNA COVID-19 vaccination series. **Methods:** A retrospective, observational, longitudinal cohort study design was used with a total eligible sample population consisting of 72,461,026 individuals in the HealthVerity dataset. The study cohort consisted of individuals ≥18 years in the United States with evidence of a COVID-19 primary vaccination series (Ad26.COV2.S or mRNA) administered between 1 January 2021 and 6 July 2022. Two exposure groups were considered based on retrospective database classification: a heterologous Ad26.COV2.S booster and a homologous Ad26.COV2.S booster. Individuals eligible for the referent groups, defined as those with a primary vaccine series alone, were identified through exact matching by age, sex, time since primary series vaccine, location, and Gagne comorbidity score. Propensity score-matched Cox proportional hazards models were used to evaluate outcomes, including COVID-19-related hospitalization and medically attended COVID-19. **Results:** Depending on the comparison group of interest, the adjusted hazard ratios for COVID-19-related hospitalization ranged from 0.63 (95% CI: 0.56, 0.72) to 0.82 (95% CI: 0.75, 0.90), and 0.93 (95% CI: 0.90, 0.96) to 0.94 (95% CI: 0.91, 0.97) for medically attended COVID-19, both favoring booster vaccination. **Conclusions:** The results of this study demonstrate the effectiveness of an Ad26.COV2.S booster vaccination compared to primary series vaccination in preventing COVID-19 hospitalization and medically attended COVID-19 for at least 12 months. This study adds to the scientific evidence that demonstrates the importance of COVID-19 booster vaccinations to support public health policy.

## 1. Introduction

The development and distribution of effective vaccines in the general population was proven to be a successful strategy for reducing severe acute respiratory syndrome coronavirus 2 (SARS-CoV-2) transmission and the burden of coronavirus disease 2019 (COVID-19) [1,2]. The Ad26.COV2.S COVID-19 vaccine, a one-dose adenovirus primary vaccination (Johnson & Johnson Innovative Medicine), was given emergency use authorization (EUA) for adults aged 18+ by the Food and Drug Administration (FDA) on 27 February 2021 [3]. Despite the uptake rate of COVID-19 vaccines (of any type) reaching greater than 50% by May 2021 [4], the emergence of more transmissible SARS-CoV-2 variants (Delta and Omicron) in the summer and early fall months led to an increase in breakthrough infections among vaccinated persons and an observed waning vaccine effectiveness [5].

In September and October of 2021, the FDA amended the EUAs for the three COVID-19 vaccines, including Ad26.COV2.S, to allow a one-dose booster vaccination (from any manufacturer) at least 2 months after the completion of the Ad26.COV2.S primary regimen or at least 6 months after the completion of a two-dose mRNA primary series for eligible individuals [6,7]. The FDA also authorized the use of heterologous booster doses, allowing for “mix-and-match” primary and booster COVID-19 vaccine regimens [8]. In preclinical and phase 2 studies, vaccine booster doses demonstrated an increase in binding and neutralizing antibodies, which are thought to increase protection against infection and severe illness [9,10]. FDA decision-making was based on data from clinical trials that demonstrated the efficacy of a homologous Ad26.COV2.S booster dose when administered 2 months after the primary vaccination and a heterologous booster dose when administered 3 months after a primary series [11,12,13].

Several studies, including a recent meta-analysis, have assessed the longer-term effectiveness of booster vaccinations [14,15]. However, evaluations of the Ad26.COV2.S COVID-19 vaccine in these studies were limited by small sample sizes, focusing on mRNA booster vaccinations or comparisons between boosted groups and unvaccinated groups. Few observational studies have reported on the effectiveness of the Ad26.COV2.S booster vaccination in heterologous and homologous combinations during the periods of Delta and Omicron variant dominance [16,17]. Both studies only included hospitalized individuals and thus do not capture the entire spectrum of individuals who have received COVID-19 booster vaccines. Evidence on the real-world relative effectiveness of homologous and heterologous Ad26.COV2.S booster vaccines over periods of high-transmissibility variants is needed to inform evolving COVID-19 public health policy globally.

Here, we report results on the effectiveness of a booster heterologous or homologous Ad26.COV2.S vaccination compared to that of a primary Ad26.COV2.S or mRNA COVID-19 vaccination series in preventing COVID-19-related hospitalization and medically attended COVID-19 using real-world data.

## 2. Materials and Methods

This was a retrospective, observational, longitudinal cohort study of the real-world effectiveness of an Ad26.COV2.S booster vaccination in preventing COVID-19-related hospitalization and medically attended COVID-19.

### 2.1. Data

The study sample was identified from the HealthVerity Marketplace, a unified database of open and closed medical and pharmacy claims, hospital transactional records for inpatient and outpatient hospital encounters, and laboratory data for individuals in the U.S. The data encompass all major payer types (commercial, Medicaid, and Medicare), and they include details on service dates, medications, diagnoses, procedures, and laboratory testing activity (including COVID-19 diagnostic and antibody tests and results). Data elements in the dataset include both open and closed claims. The HealthVerity data are nationally representative of the U.S. population across age groups and most regions, with data coverage comparable to that reported by the 2010 Census Bureau’s American Community Survey [18].

### 2.2. Population and Exposure

Adults aged 18 years or older with evidence of a COVID-19 primary series vaccination (two mRNA vaccinations from the same manufacturer [Pfizer-BioNTech or Moderna] or one Ad26.COV2.S vaccination) administered between 1 January 2021 and 6 July 2022 and with at least 365 days of medical and pharmacy pre-index enrollment were eligible for the study cohorts (Figure 1). The national drug codes (NDCs) or Current Procedural Terminology (CPT) codes used to identify the vaccines are provided in Appendix A. We considered two definitions of exposure or receipt of booster vaccination: a heterologous Ad26.COV2.S booster, defined as an Ad26.COV2.S booster administered at least 152 days after the completion of a two-dose mRNA primary series, and a homologous Ad26.COV2.S booster, defined as an Ad26.COV2.S booster administered at least 61 days after a one-dose Ad26.COV2.S primary vaccination [19]. We used 152 days because, at the time of protocol development and submission for IRB approval, the official FDA guidance of at least 6 months after the completion of a two-dose mRNA primary series for eligible individuals had not yet been developed. This 152-day period reflected the most scientifically accurate timeframe based on evolving data from clinical trials and scientific manuscripts in development. We compared groups of individuals with each exposure definition to those with an mRNA primary vaccine series (two doses) alone (i.e., without evidence of a booster vaccine) and to an Ad26.COV2.S vaccine (one dose) alone, for a total of four contrasts of interest (Table 1).

Separate analytic cohorts were identified for each contrast of interest using a combination of exact matching and propensity score (PS) matching. For each analytic cohort, individuals who received an Ad26.COV2.S booster vaccine (exposed group) were matched on the same calendar day with up to 10 referent individuals who had no evidence of a COVID-19 booster vaccination (referent group). Referent group individuals were matched by age (±5 years), sex, days since completed primary vaccination (±30 days), geographic location of residence (state) at index, and the Gagne combined comorbidity index (assessed over the 365-day baseline period) [20]. Exact matching was conducted without replacement.

After exact matching, exposed patients were then matched with up to four referent patients using parallel propensity score (PS) matching with a 1% caliper based on demographic and clinical characteristics in the baseline period in order to control for potential confounders. The PS model included the following characteristics: age, sex, Gagne comorbidity index score, calendar month of index date, calendar month of primary vaccination, state, insurance type, chronic obstructive pulmonary disease (COPD), pulmonary fibrosis, HIV infection status as defined by ICD-10-CM codes, immunocompromised status including from blood or organ transplant, liver disease, malignancies excluding non-melanoma skin cancer, moderate-to-severe asthma, cerebrovascular disease, chronic kidney disease, hypertension, serious heart condition, obesity, sickle cell disease, thalassemia, type 1 diabetes, type 2 diabetes, neurologic condition, COVID-19 history, recent (≤60 days from index) medical claims, and recent pharmacy claims. To assess the balance of measured confounders after PS matching, the absolute standardized difference (ASD) between the exposed and referent groups was examined; any measured covariates with a value <0.1 were considered balanced.

### 2.3. Follow-Up and Outcomes

COVID-19-related hospitalization and medically attended COVID-19 were assessed as study outcomes. COVID-19-related hospitalization was defined as any inpatient medical claim or inpatient hospital encounter where any medically attended COVID-19 began in the 21 days prior to the start of the hospitalization and lasted until the final day of hospitalization. Medically attended COVID-19 was defined as a positive COVID-19 diagnostic test (NAAT) or any medical claim, inpatient hospital encounter, or outpatient hospital encounter indicating a diagnosis of COVID-19 (ICD-10-CM code: U07.1). Follow-up began 14 days after the index date and ended at the receipt of any additional COVID-19 vaccine after index, death, loss to follow-up/disenrollment from the insurance plan, the end of the study period (19 October 2022), or the occurrence of the outcome of interest during the follow-up period.

### 2.4. Participant Characteristics

We evaluated the participant characteristics used in the PS matching over a 365-day baseline period using descriptive statistics. Additional baseline patient characteristics included the receipt of a COVID-19 laboratory test, the days since the primary COVID-19 vaccine series, and the number of medical and pharmacy claims. ASDs were reported for differences in characteristics between the exposed and referent groups, both pre- and post-PS matching. Demographic, clinical, and HCRU-related characteristics were described over the baseline period using summary statistics: for binary and categorical variables, counts and percentages (*n*, %) were used; for continuous variables, mean, standard deviation (SD), median, and interquartile range (IQR) were used.

### 2.5. Statistical Analysis

Incidence rates per 1000 person-years and 95% confidence intervals (CIs) were calculated for each exposure and referent group. Cox proportional hazard models were used to report hazard ratios (HRs) and 95% CIs for the relative risk of study outcomes [21]. Doubly robust outcome models were used in cases where patient characteristics remained imbalanced after PS matching [22]. The proportional hazards assumption was assessed through a visual inspection of Kaplan–Meier curves. Sensitivity analyses were conducted among individuals from the California state registry to assess the impact of potential vaccine underreporting.

To understand the potential effect of misclassification, we conducted sensitivity analyses in a subgroup of individuals residing in California using comprehensive COVID-19 vaccination data from the California state registry, linked to HealthVerity data (the results are not reported; see Appendix A). No substantial differences in the results were noted between the primary analyses and the sensitivity analyses.

## 3. Results

Of the 548,788,380 individuals in the HealthVerity dataset as of 25 August 2023, 72,461,026 had evidence of a completed primary COVID-19 vaccination between 1 January 2021 and 6 July 2022. Approximately 90% (n = 65,000,652) had an mRNA primary vaccination between 1 January 2021 and 18 March 2022 (mRNA primary vaccine eligibility period), and 10% (n = 7,158,010) had an Ad26.COV2.S primary vaccination between 1 January 2021 and 6 July 2022 (Ad26.COV2.S primary vaccine eligibility period) (Appendix A).

### 3.1. Descriptive Statistics

#### 3.1.1. Contrast: mRNA + Ad26.COV2.S vs. mRNA + No Boost

After meeting the eligibility criteria, exact matching, and PS matching, the final analytic cohort consisted of 2969 exposed patients (mRNA + Ad26.COV2.S) and 11,492 referent patients (mRNA + no boost) (Table 2). Of the 50 characteristics used for PS matching, two were unbalanced pre-PS matching (Appendix A). After PS matching, all characteristics were balanced (Table 2). In the PS-matched cohort, the mean age was 47 years in both the exposed and referent groups. In both groups, the majority were male (53.3%, exposed; 54.5%, referent), located in the west (51.4%, exposed; 51.6%, referent), and had commercial insurance (54.9%, exposed; 55.2%, referent). A plurality of patients completed their primary vaccination series in April 2021 (34.8%, both groups) and received their booster vaccine in December 2021 (39.4%, exposed; 39.2%, referent). The most common comorbidities were hypertension, neurologic conditions, and obesity. Less than 10% of the patients had a history of COVID-19 infection.

#### 3.1.2. Contrast: mRNA + Ad26.COV2.S vs. Ad26.COV2.S + No Boost

After meeting the eligibility criteria, exact matching, and PS matching, the final analytic cohort consisted of a total of 2973 exposed patients (mRNA + Ad26.COV2.S) and 11,568 referent patients (Ad26.COV2.S + no boost) (Table 2). Prior to PS matching, all characteristics were balanced (Appendix A). In the PS-matched cohort, the mean age was 47 years, and the majority of patients were male (Table 2). More than half of each group was located in the west, had commercial insurance, and received their primary vaccine in April or May 2021. The most common comorbidities were hypertension, neurologic conditions, and obesity.

#### 3.1.3. Contrast: Ad26.COV2.S + Ad26.COV2.S vs. mRNA + No Boost

After meeting the eligibility criteria, exact matching, and PS matching, the final analytic cohort consisted of 74,628 exposed patients (Ad26.COV2.S + Ad26.COV2.S) and 289,215 referent patients (mRNA + no boost) (Table 3). Of the 50 characteristics used for PS matching, one characteristic (month of primary vaccination: February 2021) was unbalanced pre-PS matching, and all characteristics were balanced post-PS matching (Appendix A). Modest differences in demographic characteristics were observed in the contrasts evaluating a homologous Ad26.COV2.S booster vaccine against the heterologous Ad26.COV2.S booster vaccine. After PS matching, the mean age was 51 years in both the exposed and referent groups. Half of the cohort was male (49.5% male in both groups). The majority of the patients were located in the western U.S. (43.4% exposed; 44.1% referent) and had commercial insurance (51.8% exposed; 52.3%). Similarly to the previous comparisons, the majority of the patients completed their primary vaccination series in April 2021 (39.7% exposed; 40.2% referent) and their booster vaccine in November 2021 (34.8% exposed and referent). The most prevalent comorbidities were hypertension, neurologic conditions, and obesity.

#### 3.1.4. Contrast: Ad26.COV2.S + Ad26.COV2.S vs. Ad26.COV2.S + No Boost

After meeting the eligibility criteria, exact matching, and PS matching, a total of 43,072 exposed patients (Ad26.COV2.S + Ad26.COV2.S) and 166,907 referent patients (Ad26.COV2.S + no boost) were included in the analytic cohort. Two characteristics (recent medical claim and recent pharmacy claim) were unbalanced pre-PS matching (Appendix A); all characteristics were balanced after PS matching (Table 3). After PS matching, the mean age was 49 years in both the exposed and referent groups. Half of the cohort was male; additionally, the cohort was predominantly located in the western U.S., and the majority had commercial insurance. The majority of the patients completed their primary vaccination series in April 2021 and their booster vaccine in November 2021. The most prevalent comorbidities were hypertension, neurologic conditions, and obesity.

### 3.2. Main Results

The incidence rate of COVID-19-related hospitalization was 12.3 hospitalizations per 1000 PY among those with an mRNA + Ad26.COV2.S exposure, compared to 18.9 hospitalizations per 1000 PY among those with an mRNA vaccine with no booster (Table 4). The adjusted HR for COVID-19-related hospitalization was 0.67 (95% CI: 0.43, 1.06). Similarly, the medically attended COVID-19 incidence rates for the exposed and referent groups were 138.2 cases and 170.7 cases per 1000 PY, respectively, resulting in an HR of 0.84 (95% CI: 0.73, 0.97).

The incidence rate of COVID-19-related hospitalization was 12.3 cases per 1000 PY among those with an mRNA + Ad26.COV2.S exposure, compared to 15.7 cases per 1000 PY among those with an Ad26.COV2.S vaccine with no booster (HR 0.81, 95% CI: 0.51, 1.27, Table 4). The HR of medically attended COVID-19 was 0.88 (95% CI: 0.76, 1.01).

Among those in the Ad26.COV2.S + Ad26.COV2.S exposure group, the incidence rate of COVID-19-related hospitalization was 10.6 per 1000 PY; in the mRNA + no boost referent group, the incidence rate was 13.9 per 1000 PY (Table 4), resulting in an HR of 0.82 (95% CI: 0.75, 0.90). A similarly higher rate of events was observed for medically attended COVID-19. The HR of medically attended COVID-19 was 0.93 (95% CI: 0.90, 0.96).

The risk of COVID-19-related hospitalization was lower among those with an Ad26.COV2.S + Ad26.COV2.S exposure than among those with an Ad26.COV2.S vaccine with no booster (HR 0.63, 95% CI: 0.56, 0.72). The HR of medically attended COVID-19 was 0.94 (95% CI: 0.91, 0.97).

We observed similar incidence rates and hazard ratios across all four contrasts of interest when limiting the study cohort to those in the California state registry (Appendix A).

## 4. Discussion

To the best of our knowledge, this is the largest study to date evaluating the real-world comparative effectiveness of heterologous and homologous booster vaccinations with Ad26.COV2.S for the prevention of COVID-19-related outcomes in the 12 months following the FDA emergency use authorization of a booster dose among a sample of U.S. individuals with a completed COVID-19 primary vaccination. Overall, the findings of this analysis demonstrate that booster vaccinations are effective in preventing COVID-19-related outcomes. The results of this study demonstrate the public health importance of boosting to prevent serious COVID-19 outcomes, including COVID-19-related hospitalization and medically attended COVID-19.

Although the Ad26.COV2.S EUA has recently expired and the vaccine is no longer available in the U.S., the findings of this study contribute to an understanding of the importance of booster vaccinations as a whole as a preventative public health measure in reducing the burden of COVID-19. Heterologous and homologous booster vaccines were effective in reducing the risk of medically attended COVID-19 by between 6% and 14%, depending on the primary vaccination type. When considering all COVID-19 booster vaccine platforms together, observational studies demonstrate higher VE against COVID-19-related hospitalization in individuals who received a booster compared to unboosted individuals when both groups (boosted and unboosted) were compared to unvaccinated individuals.

A test-negative case–control study (n = 1572 cases, n = 1609 controls) reported a VE of 77% (95% CI: 71–82%) for a primary series and booster vaccine compared to unvaccinated persons and a VE of 44% (95% CI: 31–54%; *p* < 0.001) for a primary series alone compared to unvaccinated persons [16]. Other studies of the relative effectiveness estimates of the Ad26.COV2.S booster vaccine have limited precision mainly due to their small sample sizes. A test-negative study of vaccine effectiveness (VE) among individuals hospitalized in the U.S. from 25 December 2021 to 4 April 2022, the period during which the Omicron variant was the most prevalent, reported a relative effectiveness estimate of 49% (95% CI: −9–76%) for the Ad26.COV2.S booster vaccine compared to an Ad26.COV2.S primary series vaccine alone [17]. The sample size for the number of individuals with an Ad26.COV2.S primary series was 96, and the sample size for the number of individuals with an Ad26.COV2.S primary series plus a booster was 65. A different U.S.-based test-negative VE study, which compared both individuals boosted with an Ad26.COV2.S vaccine and individuals with a primary Ad26.COV2.S vaccine alone to individuals with no COVID-19 vaccine, did not report an increased VE (Ad26.COV2.S boosted: VE 35% (95% CI: −54–73%); Ad26.COV2.S primary series only: VE 32% (95% CI: 1–54%; *p* = 0.79)) [16]. The sample size of individuals who received an Ad26.COV2.S booster dose was 25, making it difficult to derive meaningful inferences. A strength of the current study is the relatively larger sample size for both heterologous and homologous COVID-19 booster groups.

This study has several limitations. COVID-19 vaccination status may be under-ascertained. The data source leveraged for this study captures real-world vaccine status in the study sample using CPT or NDC codes in procedure and/or pharmacy claims, respectively. Not all providers in the U.S. submit for reimbursement for vaccine administration due to no-cost government vaccination programs.

Similarly, COVID-19 infection status may be under-ascertained due to the impact of at-home testing. We caution that the results of this study focus on more severe COVID-19 outcomes of interest rather than on non-serious outcomes; as a result, the incidence rates of the outcomes observed in this study may be lower than those observed in general COVID-19-related infections that may be captured in a clinical trial setting. However, the use of different data sources to ascertain outcomes (e.g., evidence of diagnoses, procedures, and lab test results) allows for improved capture of COVID-19 events.

There may be unmeasured or residual confounding in this observational study. Unlike in a clinical trial, exposure assignment was not randomized, resulting in the possibility of unmeasured confounding of an unknown magnitude of the exposure–outcome association. Disruptions to the healthcare system as a result of the COVID-19 pandemic, along with the accompanying influence of COVID-19 on disease outcomes, precluded the use of methods such as negative controls to assess for such potential confounding. However, at baseline, the exposure groups were overall similar, with two or fewer measured characteristics imbalanced prior to PS matching. To mitigate bias and facilitate confounding control, this study measured a large set of demographic, clinical, and utility-related characteristics. PS matching and doubly robust model specification were then applied to this set of characteristics, mimicking a clinical trial’s randomization and minimizing the chance of spurious effect estimates due to potential confounders.

This study used data from a large and generalizable U.S. population-based dataset. Real-world data are generated during routine clinical care and are representative of the underlying populations. The source data include 16 individual-linked data sources of medical and pharmacy claims data, hospital transactional records for inpatient and outpatient hospital encounters (also known as chargemaster data), and laboratory data, and they include populations covered by commercial insurance, Medicare Advantage, and Medicaid. The results of this study may be generalizable to the broader U.S. adult population.

## 5. Conclusions

This population-based observational cohort study from clinical practice in the U.S. demonstrates the effectiveness of an Ad26.COV2.S booster vaccination for at least 12 months. In the U.S., individuals who were boosted with Ad26.COV2.S were at a lower risk of COVID-19 hospitalization and medically attended COVID-19 than unboosted individuals. This study adds to the volume of evidence that demonstrates the added benefit of COVID-19 vaccine booster doses that is critical to guiding the development of sound public health policy.

## Figures and Tables

**Figure 1 vaccines-13-00166-f001:**
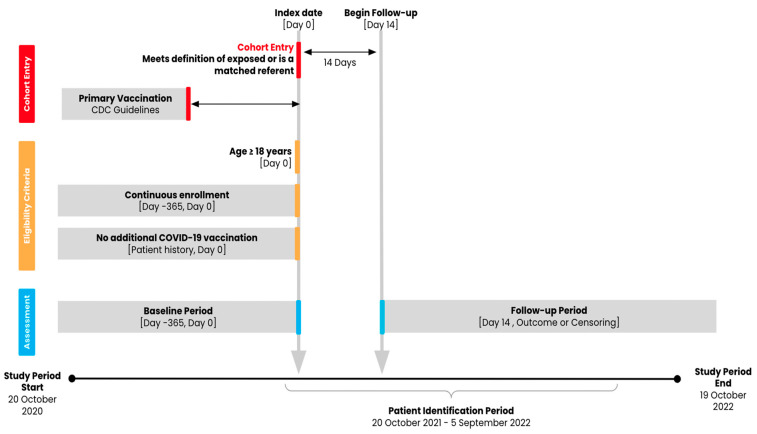
Study design diagram.

**Table 1 vaccines-13-00166-t001:** Exposure contrasts of interest.

	Contrasts of Interest	
	Exposure	Referent
1	2 mRNA + 1 Ad26.COV2.S	2 mRNA
2	2 mRNA + 1 Ad26.COV2.S	1 Ad26.COV2.S
3	1 Ad26.COV2.S + 1 Ad26.COV2.S	2 mRNA
4	1 Ad26.COV2.S + 1 Ad26.COV2.S	1 Ad26.COV2.S

“+” differentiates between the primary vaccination series and booster vaccination such that “2 mRNA + 1 Ad26.COV2.S” refers to a two-dose mRNA primary vaccination series followed by a single-dose Ad26.COV2.S booster vaccine. mRNA refers to a BNT162b2 or mRNA-1273 vaccination. The completed primary vaccination series (i.e., 2 mRNA vaccinations) must be from the same manufacturer (i.e., homologous primary vaccination). “Referent” is defined as the comparator group consisting of individuals with an mRNA primary vaccine series (two doses) alone (i.e., without evidence of a booster vaccine) and those with an Ad26.COV2.S vaccine (one dose) alone.

**Table 2 vaccines-13-00166-t002:** Baseline characteristics, post-PS matching, for mRNA + Ad26.COV2.S vs. primary series.

Characteristic ^a^	mRNA + Ad26.COV2.S vs. mRNA + No Boost			mRNA + Ad26.COV2.S vs. Ad26.COV2.S + No Boost		
N (%) or Mean +/− SD Unless Otherwise Noted	mRNA + Ad26.COV2.S	mRNA + No Boost	ASD	mRNA + Ad26.COV2.S	Ad26.COV2.S + No Boost	ASD
Number of individuals	2969	11,492		2973	11,568	
Demographics						
Age, mean (SD)	46.89 (15.58)	46.69 (15.29)	0.013	46.89 (15.56)	46.88 (15.40)	0.000
Male sex; n (%)	1581 (53.3%)	6253 (54.4%)	0.023	1587 (53.4%)	6170 (53.3%)	0.001
U.S. region ^b^			0.005			0.003
Northeast; n (%)	474 (16.0%)	1820 (15.8%)		474 (15.9%)	1836 (15.9%)	
Midwest; n (%)	320 (10.8%)	1229 (10.7%)		321 (10.8%)	1242 (10.7%)	
South; n (%)	648 (21.8%)	2511 (21.8%)		651 (21.9%)	2537 (21.9%)	
West; n (%)	1527 (51.4%)	5932 (51.6%)		1527 (51.4%)	5953 (51.5%)	
State ^c^			0.053			0.042
Index months			0.020			0.018
October 2021; n (%)	125 (4.2%)	499 (4.3%)		125 (4.2%)	509 (4.4%)	
November 2021; n (%)	877 (29.5%)	3343 (29.1%)		876 (29.5%)	3360 (29.0%)	
December 2021; n (%)	1169 (39.4%)	4502 (39.2%)		1171 (39.4%)	4567 (39.5%)	
January 2022; n (%)	403 (13.6%)	1609 (14.0%)		406 (13.7%)	1597 (13.8%)	
February 2022 to September 2022; n (%)	395 (13.3%)	1539 (13.4%)		395 (13.3%)	1535 (13.3%)	
Commercial Enrollment on CED; n (%)	1631 (54.9%)	6343 (55.2%)	0.005	1634 (55.0%)	6338 (54.8%)	0.003
Medicare Advantage Enrollment on CED; n (%)	203 (6.8%)	795 (6.9%)	0.003	201 (6.8%)	783 (6.8%)	0.000
Medicaid Enrollment on CED; n (%)	1245 (41.9%)	4762 (41.4%)	0.010	1247 (41.9%)	4845 (41.9%)	0.001
*COVID-19-related characteristics*						
Receipt of any laboratory test for COVID-19; n (%)	832 (28.0%)	3214 (28.0%)	0.001	835 (28.1%)	3294 (28.5%)	0.009
History of COVID-19 infection; n (%)	269 (9.1%)	1024 (8.9%)	0.005	270 (9.1%)	1052 (9.1%)	0.000
Month of primary vaccination; n (%) ^d^						
January 2021; n (%)	91 (3.1%)	328 (2.9%)	0.012	91 (3.1%)	377 (3.3%)	0.011
February 2021; n (%)	301 (10.1%)	1150 (10.0%)	0.004	301 (10.1%)	1150 (9.9%)	0.006
March 2021; n (%)	392 (13.2%)	1502 (13.1%)	0.004	392 (13.2%)	1549 (13.4%)	0.006
April 2021; n (%)	1033 (34.8%)	3997 (34.8%)	0.000	1032 (34.7%)	3949 (34.1%)	0.012
May 2021; n (%)	752 (25.3%)	2947 (25.6%)	0.007	755 (25.4%)	2910 (25.2%)	0.006
June 2021 to July 2022; n (%)	400 (13.5%)	1568 (13.6%)	-	402 (13.5%)	1633 (14.1%)	-
Days since primary series (SD) ^a^	245.07 (52.94)	244.93 (54.54)	0.003	245.02 (52.81)	244.88 (53.81)	0.002
*Comorbid conditions*						
Cerebrovascular disease; n (%)	101 (3.4%)	385 (3.4%)	0.003	101 (3.4%)	358 (3.1%)	0.017
Chronic kidney disease (CKD); n (%)	122 (4.1%)	462 (4.0%)	0.005	122 (4.1%)	472 (4.1%)	0.001
Chronic obstructive pulmonary disease (COPD); n (%)	397 (13.4%)	1544 (13.4%)	0.002	399 (13.4%)	1528 (13.2%)	0.006
Cystic fibrosis; n (%)	0 (0.0%)	0 (0.0%)	-	0 (0.0%)	0 (0.0%)	-
HIV; n (%)	33 (1.1%)	126 (1.1%)	0.001	32 (1.1%)	133 (1.1%)	0.007
Hypertension; n (%)	923 (31.1%)	3558 (31.0%)	0.003	922 (31.0%)	3590 (31.0%)	0.000
Immunocompromised state from organ transplant; n (%)	10 (0.3%)	40 (0.3%)	0.002	10 (0.3%)	41 (0.4%)	0.003
Immunocompromised state from blood transplant; n (%)	13 (0.4%)	41 (0.4%)	0.013	13 (0.4%)	56 (0.5%)	0.007
Liver disease; n (%)	205 (6.9%)	766 (6.7%)	0.010	207 (7.0%)	808 (7.0%)	0.001
Malignancies; n (%)	105 (3.5%)	393 (3.4%)	0.006	105 (3.5%)	432 (3.7%)	0.011
Moderate-to-severe asthma; n (%)	35 (1.2%)	142 (1.2%)	0.005	35 (1.2%)	136 (1.2%)	0.000
Neurologic conditions; n (%)	950 (32.0%)	3626 (31.6%)	0.010	954 (32.1%)	3651 (31.6%)	0.011
Obesity; n (%)	730 (24.6%)	2792 (24.3%)	0.007	730 (24.6%)	2829 (24.5%)	0.002
Pulmonary fibrosis; n (%)	12 (0.4%)	48 (0.4%)	0.002	12 (0.4%)	44 (0.4%)	0.004
Serious heart conditions; n (%)	258 (8.7%)	998 (8.7%)	0.000	259 (8.7%)	1003 (8.7%)	0.001
Sickle cell disease; n (%)	0 (0.0%)	0 (0.0%)	-	0 (0.0%)	0 (0.0%)	-
Thalassemia; n (%)	2 (0.1%)	4 (0.0%)	0.014	2 (0.1%)	7 (0.1%)	0.003
Type 1 diabetes mellitus; n (%)	38 (1.3%)	156 (1.4%)	0.007	38 (1.3%)	155 (1.3%)	0.005
Type 2 diabetes mellitus; n (%)	446 (15.0%)	1713 (14.9%)	0.003	444 (14.9%)	1714 (14.8%)	0.003
Gagne combined comorbidity score, mean (SD)	0.80 (1.76)	0.79 (1.75)	0.009	0.81 (1.76)	0.79 (1.75)	0.008
*Healthcare resource utilization*						
Recent medical claim; n (%) ^e^	2017 (67.9%)	7640 (66.5%)	0.031	2018 (67.9%)	7727 (66.8%)	0.023
Recent pharmacy claim; n (%) ^e^	2058 (69.3%)	7811 (68.0%)	0.029	2061 (69.3%)	7982 (69.0%)	0.007

Abbreviations: ASD = absolute standardized difference; CKD = chronic kidney disease; COPD = chronic obstructive pulmonary disease; HIV = human immunodeficiency virus; ICU = intensive care unit; Ad26.COV2.S = Johnson & Johnson; mRNA = messenger ribonucleic acid; PS = propensity score; SD = standard deviation. ^a^ Characteristics reported for population matched by propensity scores. Unless otherwise noted, demographic variables were assessed at cohort entry (index), and comorbidities and clinical utilization variables were assessed during the 1 year before cohort entry. ^b^ Variables excluded from the final propensity score models for overall and all stratified results. ^c^ Individual state covariates are not shown for brevity. ^d^ Month of primary vaccination refers to the month of completion of either a single-dose Ad26.COV2.S primary vaccine series occurring at any time between 1 January 2021 and 6 July 2022 (61 days prior to the last possible index date) or a homologous two-dose mRNA primary vaccine series occurring between 1 January 2021 and 6 April 2022 (152 days prior to the last possible index date). June 2021–July 2022 are presented as aggregated frequencies; ASDs are not reported. ^e^ Recent medical and pharmacy claims were defined as claims beginning during the 60 days before cohort entry.

**Table 3 vaccines-13-00166-t003:** Baseline characteristics, post-PS matching, for Ad26.COV2.S + Ad26.COV2.S vs. primary series.

Characteristic ^a^	Ad26.COV2.S + Ad26.COV2.S vs. mRNA + No Boost			Ad26.COV2.S + Ad26.COV2.S vs. Ad26.COV2.S + No Boost		
N (%) or Mean +/− SD Unless Otherwise Noted	Ad26.COV2.S + Ad26.COV2.S	mRNA + No Boost	ASD	Ad26.COV2.S + Ad26.COV2.S	Ad26.COV2.S + No Boost	ASD
Number of individuals	74,628	289,215		43,072	166,907	
Demographics						
Age, mean (SD)	51.19 (14.93)	51.14 (14.69)	0.003	49.08 (15.46)	49.04 (15.03)	0.003
Male sex; n (%)	36,904 (49.5%)	143,114 (49.5%)	0.001	21,373 (49.6%)	82,901 (49.7%)	0.001
U.S. region ^b^			0.014			0.014
Northeast; n (%)	14,866 (19.9%)	56,825 (19.6%)		8838 (20.5%)	33,913 (20.3%)	
Midwest; n (%)	12,125 (16.2%)	46,705 (16.1%)		7593 (17.6%)	29,046 (17.4%)	
South; n (%)	15,255 (20.4%)	58,243 (20.1%)		9536 (22.1%)	36,558 (21.9%)	
West; n (%)	32,382 (43.4%)	127,442 (44.1%)		17,105 (39.7%)	67,390 (40.4%)	
State ^c^			0.021			0.022
Index months			0.008			0.008
October 2021; n (%)	5981 (8.0%)	22,810 (7.9%)		5593 (13.0%)	21,758 (13.0%)	
November 2021; n (%)	25,936 (34.8%)	100,700 (34.8%)		17,467 (40.6%)	68,010 (40.7%)	
December 2021; n (%)	22,071 (29.6%)	86,077 (29.8%)		10,297 (23.9%)	39,928 (23.9%)	
January 2022; n (%)	12,009 (16.1%)	46,440 (16.1%)		5457 (12.7%)	20,981 (12.6%)	
February 2022 to September 2022; n (%)	8631 (11.6%)	33,188 (11.5%)		4258 (9.9%)	16,230 (9.7%)	
Commercial Enrollment on CED; n (%)	38,657 (51.8%)	151,204 (52.3%)	0.010	22,411 (52.0%)	87,390 (52.4%)	0.007
Medicare Advantage Enrollment on CED; n (%)	6748 (9.0%)	25,518 (8.8%)	0.008	3593 (8.3%)	13,676 (8.2%)	0.005
Medicaid Enrollment on CED; n (%)	30,444 (40.8%)	117,236 (40.5%)	0.005	17,825 (41.4%)	68,698 (41.2%)	0.005
*COVID-19-related characteristics*						
Receipt of any laboratory test for COVID-19; n (%)	16,466 (22.1%)	63,213 (21.9%)	0.005	9619 (22.3%)	36,575 (21.9%)	0.010
History of COVID-19 infection; n (%)	6371 (8.5%)	24,115 (8.3%)	0.007	3724 (8.6%)	14,086 (8.4%)	0.007
Month of primary vaccination; n (%) ^d^						
January 2021; n (%)	0 (0.0%)	0 (0.0%)	-	0 (0.0%)	0 (0.0%)	-
February 2021; n (%)	1 (0.0%)	4 (0.0%)	0.000	0 (0.0%)	0 (0.0%)	-
March 2021; n (%)	20,103 (26.9%)	77,512 (26.8%)	0.003	10,407 (24.2%)	40,411 (24.2%)	0.001
April 2021; n (%)	29,642 (39.7%)	116,132 (40.2%)	0.009	15,881 (36.9%)	62,149 (37.2%)	0.008
May 2021; n (%)	11,169 (15.0%)	42,657 (14.7%)	0.006	6933 (16.1%)	26,638 (16.0%)	0.004
June 2021 to July 2022; n (%)	13,713 (18.4%)	52,910 (18.3%)	-	9851 (22.9%)	37,709 (22.6%)	-
Days since primary series (SD) ^a^	230.89 (52.73)	228.19 (52.57)	0.051	215.99 (51.55)	216.08 (51.60)	0.002
*Comorbid conditions*						
Cerebrovascular disease; n (%)	2599 (3.5%)	9786 (3.4%)	0.005	1438 (3.3%)	5475 (3.3%)	0.003
Chronic kidney disease (CKD); n (%)	3824 (5.1%)	14,510 (5.0%)	0.005	1997 (4.6%)	7642 (4.6%)	0.003
Chronic obstructive pulmonary disease (COPD); n (%)	9521 (12.8%)	36,249 (12.5%)	0.007	5460 (12.7%)	20,861 (12.5%)	0.005
Cystic fibrosis; n (%)	2 (0.0%)	8 (0.0%)	0.000	1 (0.0%)	4 (0.0%)	0.000
HIV; n (%)	541 (0.7%)	2110 (0.7%)	0.001	313 (0.7%)	1189 (0.7%)	0.002
Hypertension; n (%)	27,665 (37.1%)	105,077 (36.3%)	0.015	14,901 (34.6%)	56,873 (34.1%)	0.011
Immunocompromised state from organ transplant; n (%)	387 (0.5%)	1504 (0.5%)	0.000	226 (0.5%)	859 (0.5%)	0.001
Immunocompromised state from blood transplant; n (%)	398 (0.5%)	1494 (0.5%)	0.002	211 (0.5%)	809 (0.5%)	0.001
Liver disease; n (%)	4850 (6.5%)	18,217 (6.3%)	0.008	2713 (6.3%)	10,232 (6.1%)	0.007
Malignancies; n (%)	3016 (4.0%)	11,496 (4.0%)	0.003	1640 (3.8%)	6230 (3.7%)	0.004
Moderate-to-severe asthma; n (%)	1013 (1.4%)	3851 (1.3%)	0.002	605 (1.4%)	2351 (1.4%)	0.000
Neurologic conditions; n (%)	23,443 (31.4%)	88,880 (30.7%)	0.015	13,303 (30.9%)	50,507 (30.3%)	0.014
Obesity; n (%)	19,819 (26.6%)	75,199 (26.0%)	0.013	11,117 (25.8%)	42,301 (25.3%)	0.011
Pulmonary fibrosis; n (%)	442 (0.6%)	1663 (0.6%)	0.002	246 (0.6%)	946 (0.6%)	0.001
Serious heart conditions; n (%)	6927 (9.3%)	26,164 (9.0%)	0.008	3697 (8.6%)	14,064 (8.4%)	0.006
Sickle cell disease; n (%)	67 (0.1%)	253 (0.1%)	0.001	43 (0.1%)	168 (0.1%)	0.000
Thalassemia; n (%)	96 (0.1%)	357 (0.1%)	0.001	52 (0.1%)	185 (0.1%)	0.003
Type 1 diabetes mellitus; n (%)	831 (1.1%)	3118 (1.1%)	0.003	457 (1.1%)	1740 (1.0%)	0.002
Type 2 diabetes mellitus; n (%)	13,604 (18.2%)	51,170 (17.7%)	0.014	7146 (16.6%)	27,406 (16.4%)	0.005
Gagne combined comorbidity score, mean (SD)	0.69 (1.66)	0.67 (1.66)	0.012	0.68 (1.64)	0.67 (1.64)	0.008
*Healthcare resource utilization*						
Recent medical claim; n (%) ^e^	48,085 (64.4%)	183,968 (63.6%)	0.017	27,622 (64.1%)	106,136 (63.6%)	0.011
Recent pharmacy claim; n (%) ^e^	53,039 (71.1%)	202,604 (70.1%)	0.022	30,194 (70.1%)	115,927 (69.5%)	0.014

Abbreviations: ASD = absolute standardized difference; CKD = chronic kidney disease; COPD = chronic obstructive pulmonary disease; HIV = human immunodeficiency virus; ICU = intensive care unit; Ad26.COV2.S = Johnson & Johnson; mRNA = messenger ribonucleic acid; PS = propensity score; SD = standard deviation. ^a^ Characteristics reported for population matched by propensity scores. Unless otherwise noted, demographic variables were assessed at cohort entry (index), and comorbidities and clinical utilization variables were assessed during the 1 year before cohort entry. ^b^ Variables excluded from the final propensity score models for overall and all stratified results. ^c^ Individual state covariates are not shown for brevity. ^d^ Month of primary vaccination refers to the month of completion of either a single-dose Ad26.COV2.S primary vaccine series occurring at any time between 1 January 2021 and 6 July 2022 (61 days prior to the last possible index date) or a homologous two-dose mRNA primary vaccine series occurring between 1 January 2021 and 6 April 2022 (152 days prior to the last possible index date). June 2021–July 2022 are presented as aggregated frequencies; ASDs are not reported. ^e^ Recent medical and pharmacy claims were defined as claims beginning during the 60 days before cohort entry.

**Table 4 vaccines-13-00166-t004:** Adjusted hazard ratios for COVID-19-related hospitalization and medically attended COVID-19.

	N Events	Person-Years	Incidence Rate (per 1000 Person-Years)	Median Days Follow-Up [IQR]	Fully Adjusted HR (95% CI) *
mRNA + Ad26.COV2.S vs. mRNA + no boost					
COVID-19-related hospitalization					
mRNA + Ad26.COV2.S	23	1867	12.32	272 (168, 303)	0.67 (0.43, 1.06)
mRNA + no boost (ref)	103	5464	18.85	208 (35, 295)	
Medically attended COVID-19					
mRNA + Ad26.COV2.S	246	1780	138.21	263 (147, 301)	0.84 (0.73, 0.97)
mRNA + no boost (ref)	876	5133	170.67	176 (30, 289)	
mRNA + Ad26.COV2.S vs. Ad26.COV2.S + no boost					
COVID-19-related hospitalization					
mRNA + Ad26.COV2.S	23	1868	12.31	272 (168, 303)	0.81 (0.51, 1.27)
Ad26.COV2.S + no boost (ref)	93	5916	15.72	226 (56, 296)	
Medically attended COVID-19					
mRNA + Ad26.COV2.S	247	1781	138.67	263 (147, 301)	0.88 (0.76, 1.01)
Ad26.COV2.S + no boost (ref)	899	5568	161.46	197 (45, 292)	
Ad26.COV2.S + Ad26.COV2.S vs. mRNA + no boost					
COVID-19-related hospitalization					
Ad26.COV2.S + Ad26.COV2.S	553	51,428	10.75	282 (219, 312)	0.82 (0.75, 0.90)
mRNA + no boost (ref)	1916	137,465	13.94	215 (33, 294)	
Medically attended COVID-19					
Ad26.COV2.S + Ad26.COV2.S	6389	48,792	130.94	273 (188, 309)	0.93 (0.90, 0.96)
mRNA + no boost (ref)	19,237	129,833	148.17	179 (30, 287)	
Ad26.COV2.S + Ad26.COV2.S vs. Ad26.COV2.S + no boost					
COVID-19-related hospitalization					
Ad26.COV2.S + Ad26.COV2.S	315	30,038	10.49	292 (217, 321)	0.63 (0.56, 0.72)
Ad26.COV2.S + no boost (ref)	1534	88,825	17.27	251 (47, 308)	
Medically attended COVID-19					
Ad26.COV2.S + Ad26.COV2.S	3813	28,427	134.13	279 (184, 318)	0.94 (0.91, 0.97)
Ad26.COV2.S + no boost (ref)	12,425	83,587	148.65	226 (41, 304)	

Abbreviations: CI = confidence interval; HR = hazard ratio; IQR = interquartile range; Ad26.COV2.S = Johnson & Johnson; mRNA = messenger ribonucleic acid. * Fully adjusted hazard ratio refers to the hazard ratio obtained from a Cox proportional hazards model comparing study groups that were balanced using PS matching.

## Data Availability

Restrictions apply to the availability of these data. Data were obtained from HealthVerity and are available from the authors with the permission of HealthVerity.

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
