# Peer review of "Effectiveness of Heterologous and Homologous Ad26.COV2.S Vaccine Boosting in Preventing COVID-19-Related Outcomes Among Individuals with a Completed Primary Vaccination Series in the United States"

_vaccines, 2025, doi:10.3390/vaccines13020166_

Round 1

Reviewer 1 Report

Comments and Suggestions for Authors

I was invited to revise the paper entitled "Effectiveness of Heterologous and Homologous Ad26.COV2.S Vaccine Boosting to Prevent COVID-19-Related Outcomes Among Individuals with a Completed Primary Vaccination Series in the United States". It was a cohort study based on real world data aimed to evaluate the effectiveness of a booster heterologous or homologous Ad26.COV2.S vaccination compared to that of a primary Ad26.COV2.S or mRNA COVID19 vaccination series in preventing COVID-19-related hospitalization and medically attended COVID-19.

The study was interesting and it was focused on a relevant topic for public health.

About methods, authors compared 4 different exposures:

1 2 mRNA + 1 Ad26.COV2.S vs 2 mRNA

2 2 mRNA + 1 Ad26.COV2.S vs 1 Ad26.COV2.S

3 1 Ad26.COV2.S + 1 Ad26.COV2.S 2 vs mRNA

4 1 Ad26.COV2.S + 1 Ad26.COV2.S vs 1 Ad26.COV2.S 

Authors performed a propensity score matching procedure for baseline characteristics in order obtain comparable study groups for baseline characteristics.

Observations:

The methodology appears robust and results are clearly presented.

- Authors should clarify in methods and under the table 4 if propensity score was used in the adjustment procedure;

- Discussions reported poor references. No compare with previous literature was perdormed. Authors should also discuss the efficacy of  Ad26.COV2.S with other covid-19 vaccines presented in other similar studies.

Author Response

Reviewer 1 Comment 1: Authors should clarify in methods and under the table 4 if propensity score was used in the adjustment procedure.

Author Response: We thank the reviewer for the comment. Propensity score matching was used and defined in the Methods section under sections 2.2 (Population and Exposure) and 2.5 (Statistical Analysis). In addition, the use of propensity score matching is indicated in the footer of Table 4.

Reviewer 1 Comment 2: Discussions reported poor references. No compare with previous literature was performed. Authors should also discuss the efficacy of  Ad26.COV2.S with other covid-19 vaccines presented in other similar studies.

Author Response: We thank the reviewer for this comment. The authors would like to note that a thorough review of the literature was conducted to identify previous studies that could be included as part of the discussion with the study’s findings. However, there is a scarcity of powered studies that have specifically explored the effectiveness of Ad26.COV2.S when used as a homologous booster (2 doses of Ad26.COV2.S) or a heterologous booster (a single dose of Ad26.COV2.S plus an additional mRNA vaccine dose).

Reviewer 2 Report

Comments and Suggestions for Authors

Nyaku et al studied the efficacy of COVID-19 booster vaccine Ad26.Cov2 in a large population. They showed  that booster vaccine is effective to prevent SARS-CoV2 infection for newer variants. These results have significance for using booster vaccine for COVID-19.

This study has following concerns

1.       The manuscript does not have line numbers, so difficult to pinpoint the concerns.

2.       In introduction, authors should define homologous and heterologous booster dose.

3.       In introduction (2nd paragraph), authors wrote that FD approved ad26.Cov2.s at least after 6 months of two doses of primary vaccines. In methods, they wrote for heterologous booster, at least 152 days after two doses of primary vaccination! Why authors took 152 days instead of at least 180 days?

4.       Table 1, Referent group should be defined in the table legend. Authors should explain rows 2, 3, and 4. Row 1 is understood but other rows are confusing, e.g in row 2, if 1 dose of AD26 is referent, then how there could be 2 mRNA (+ 1 AD26)?

5.       In Table 4, after 2mRNA vaccine with no boost, the percentages of COVID hospitalization + medical attention covers (1.8%+17%=18.8% people were affected. Similarly for AD26, with no boosts (1.727% +14.8%) =16.5% were affected by both hospitalization and medical attention. Are the numbers very high after initial vaccination? Again these percentages are not just affected, only those severely ill, thus much more were infected and not tested. So how many percent of people got infected after complete  doses of primary vaccines? Does these numbers explain vaccines efficacy at all? Authors should add comments on these in discussion.

6.       In conclusion, authors wrote that booster doses have substantial benefit in guiding development of sound public health policy against COVID-19. Actually the differences between booster vs non-booster are not very prominent (for 2mRNA/AD26, 1.23% vs 1.88%; for AD26/AD26, 1.049% vs 1.72%) although hazard ratio goes down.  The differences for medical attention are further negligible (for 2mRNA/AD26, 13.8% vs 17.0 or for AD26/AD26, 13.4% vs 14.8%). Furthermore, when the study was done, omicron lost its power of virulency than alpha or delta.  Are these contributions of the booster are very significant? Authors should comment on discussion.

Author Response

Reviewer 2 Comment 1:  The manuscript does not have line numbers, so difficult to pinpoint the concerns.

Author Response: The authors used a manuscript template that is authorized by Vaccines MDPI which unfortunately does not include line numbers.

Reviewer 2 Comment 2: In introduction, authors should define homologous and heterologous booster dose.

Author Response: We thank the reviewer for this comment, however since definitions of a study are inherently tied to how a study was executed, the authors feel the appropriate section for definitions is under "Materials and Methods". That said, homologous and heterologous booster dose definitions are outlined in Section 2.2 (Population and Exposure) under the "Materials and Methods" section.

Reviewer 2 Comment 3:  In introduction (2nd paragraph), authors wrote that FDA approved Ad26.COV2.S at least after 6 months of two doses of primary vaccines. In methods, they wrote for heterologous booster, at least 152 days after two doses of primary vaccination! Why authors took 152 days instead of at least 180 days?

Author Response: We thank the reviewer for this comment. At the time of protocol development/approval in 2021, specific guidance around the period from receipt of primary COVID-19 vaccines to receiving a booster dose was rapidly evolving and not definitive. We chose 152 days at the time since this reflected the most scientifically accurate timeframe based on evolving data from clinical trials and scientific manuscripts in development.

Reviewer 2 Comment 4: Table 1, Referent group should be defined in the table legend. Authors should explain rows 2, 3, and 4. Row 1 is understood but other rows are confusing, e.g in row 2, if 1 dose of AD26 is referent, then how there could be 2 mRNA (+ 1 AD26)?

Author Response: The footer of Table 1 has been updated to include a definition of "referent". Rows 2, 3, and 4 are defined upon the first mention of "Table 1" in Section 2.2 (Population and Exposure), under "Materials and Methods". In summary the primary mRNA vaccine series constituted 2 doses, while the primary vaccine series for Ad26.COV2.S, constituted a single dose.

Reviewer 2 Comment 5:   In Table 4, after 2mRNA vaccine with no boost, the percentages of COVID hospitalization + medical attention covers (1.8%+17%=18.8% people were affected. Similarly for AD26, with no boosts (1.727% +14.8%) =16.5% were affected by both hospitalization and medical attention. Are the numbers very high after initial vaccination? Again these percentages are not just affected, only those severely ill, thus much more were infected and not tested. So how many percent of people got infected after complete  doses of primary vaccines? Does these numbers explain vaccines efficacy at all? Authors should add comments on these in discussion.

Author Response: We thank the reviewer for this question. Data from HealthVerity which is cross-sectional in nature was used in conducting this analysis thus, we are unable to accurately assess incidence rates multiple times after initial vaccination. Also, since our research question specifically focused on hospitalized and medically attended patients, we are unable to comment on infected individuals not tested (and therefore not meeting the inclusion criteria for hospitalization and or medically attended). To the best of our knowledge, the estimates we've presented in this paper are accurate reflections of vaccine effectiveness (and not efficacy) following the receipt of homologous or heterologous COVID-19 vaccine booster doses.

Reviewer 2 Comment 6:    In conclusion, authors wrote that booster doses have substantial benefit in guiding development of sound public health policy against COVID-19. Actually the differences between booster vs non-booster are not very prominent (for 2mRNA/AD26, 1.23% vs 1.88%; for AD26/AD26, 1.049% vs 1.72%) although hazard ratio goes down.  The differences for medical attention are further negligible (for 2mRNA/AD26, 13.8% vs 17.0 or for AD26/AD26, 13.4% vs 14.8%). Furthermore, when the study was done, omicron lost its power of virulency than alpha or delta.  Are these contributions of the booster are very significant? Authors should comment on discussion.

Author Response: We thank the reviewer to this comment. It is worth pointing out that in a then rapidly evolving COVID-19 pandemic setting that resulted in over 100 million confirmed cases and over 1.2 million confirmed deaths in the U.S. alone, any efforts that resulted in incrementally reducing the morbidity and mortality associated with COVID-19 were hugely beneficial for public health including reducing healthcare resource utilization. The evidence provided in this manuscript suggest that booster vaccine doses could and did incrementally provide public health benefits. That said, the authors feel that the contributions of booster vaccine doses are merited. With regards to circulating COVID-19 variants, the study did not have this as a scientific research question or objective, thus we are unable to comment on this.

Reviewer 3 Report

Comments and Suggestions for Authors

The authors belong to the Data Science and Digital Health group of Johnson & Johnson Innnovative Medicine; which are the producers of the Ad26.COV2.S vaccine.  They sought to test the effectiveness of heterologous and homologous vaccine boosting to prevent COVID-19 related outcomes in patients who had completed their primary vaccination series in the United States.  They used data from HealthVerity, which is a data gathering enterprise that curates health care information from insurance companies and health care enterprises.  The data is curated and deidentified.  The source data include 16 individual-linked data sources of medical and pharmacy claims data, hospital transactional records for inpatient and out-patient hospital encounters (also known as chargemaster data), and laboratory data, and they include populations covered by commercial insurance, Medicare Advantage, and Medicaid.

The authors performed a retrospective, observational, longitudinal cohort study design was used with a total eligible sample population consisting of 72,461,026 individuals in the HealthVerity dataset.  This study cohort consisted of individuals ≥18 years in the United States with evidence of a COVID-19 primary vaccination series (Ad26.COV2.S or mRNA) administered between 1 January 2021 and 6 July 2022.  For each analytic cohort, individuals who received an Ad26.COV2.S booster vaccine (exposed group) were matched on the same calendar day with up to 10 referent individuals who had no evidence of a COVID-19 booster vaccination (referent group).  Referent group individuals were matched by age (±5 years), sex, days since completed primary vaccination (±30 days), geographic location of residence (state) at index, and the Gagne combined comorbidity index (assessed over the 365-day baseline period).  Two exposure groups were considered based on retrospective database classification: a heterologous Ad26.COV2.S booster and a homologous Ad26.COV2.S booster.  Incidence rates per 1000 person-years and 95% confidence intervals (CIs) were calculated for each exposure and referent group.  Propensity score-matched Cox proportional hazards models were used to evaluate outcomes, including COVID-19-related hospitalization and medically attended COVID-19. 

COVID-19-related hospitalization and medically attended COVID-19 were assessed as study outcomes.  The incidence rate of COVID-19-related hospitalization was 12.3 hospitalizations per 1000 PY among those with an mRNA + Ad26.COV2.S exposure, compared to 18.9 hospitalizations per 1000 PY among those with an mRNA vaccine with no booster.  The adjusted HR for COVID-19-related hospitalization was 0.67 (95% CI: 0.43, 1.06).  The medically attended COVID-19 incidence rates for the exposed and referent groups were 138.2 cases and 170.7 cases per 1000 PY, respectively, resulting in an HR of 0.84 (95% CI: 0.73, 0.97).  Similar benefits were observed for the three other contrast groups.  Essentially, heterologous and homologous booster vaccines were effective in reducing the risk of medically attended COVID-19 between 6% and 14%, depending on the primary vaccination type.

In the discussion, the data above was compared to several limited studies of vaccine efficacy in general.  This comparison took up only one paragraph.  Interestingly, one of the studies reported no increased vaccine efficacy after vaccination. 

There was also a limited discussion of potential limitations.

The authors concluded that In the U.S., individuals who were boosted with Ad26.COV2.S were at a lower risk of COVID-19 hospitalization and medically attended COVID-19 than unboosted individuals. 

Points

The issue of vaccine efficacy for COVID-19 remains a hot issue, and is likely to remain so for the next generation.  This includes both medical professionals and the population in general, many of whom are still asking the question of what exactly happened in the vaccine response to COVID.  The authors themselves point out that the studies of vaccine efficacy that they identified in the literature were limited in scope and provided conflicting evidence of efficacy.  The authors’ use of HealthVerity information, which is tremendous in number, provides an exceptional opportunity for clarification of this issue, which is raised by the authors but not addressed from their own data.

1.      The key to any epidemiological study is the definition of the population to be evaluated, the inclusion and exclusion criteria, and a detailed analysis of the limitations of the dataset.  In this case, the discussion of the pluses and minuses of the HealthVerity dataset is very superficial. 

a.      Just one example, the title of the paper refers to an analysis of vaccine effectiveness in the United States.  The results section begins with the statement that there were 548,788,380 individuals in the HealthVerity dataset as of 25 August 2023.  The population of the United States was only 334,233, 854 on 1 Jan 2023, according to the US census bureau.  Where did the 214 million additional individuals in the database come from?  It is likely that there were repeated instances of the same individuals, but how these were compiled and the dataset corrected is not discussed.

2.      It has been repeatedly reported that the beneficial effects of the vaccine are age related.  The older you are, the greater the benefit.  Did they ever stratify their results on the basis of age?

3.      Again, vaccine effectiveness is the key issue, and the authors have stated that the literature available to them is conflicting and limited in scope.  Their data can provide a general answer to the question of vaccine efficacy in a large proportion of the medical records from a substantial segment of the people of the United States.  Only with information as background can we properly interpret the significance of their findings regarding booster vaccine efficacy. 

Author Response

Reviewer 3 Comment 1: The key to any epidemiological study is the definition of the population to be evaluated, the inclusion and exclusion criteria, and a detailed analysis of the limitations of the dataset.  In this case, the discussion of the pluses and minuses of the HealthVerity dataset is very superficial. 

  1. Just one example, the title of the paper refers to an analysis of vaccine effectiveness in the United States.  The results section begins with the statement that there were 548,788,380 individuals in the HealthVerity dataset as of 25 August 2023.  The population of the United States was only 334,233, 854 on 1 Jan 2023, according to the US census bureau.  Where did the 214 million additional individuals in the database come from?  It is likely that there were repeated instances of the same individuals, but how these were compiled and the dataset corrected is not discussed.

 Author Response: We agree with the reviewer about specific characteristics that strengthen an epidemiologic study. Within, the three paragraphs of the "Discussion" section dedicated to study limitations, we've outlined  and explained specific attributes of the data source (HealthVerity) that we've identified as limitations, e.g., disease under-ascertainment, unmeasured or residual confounding and measures adopted to minimize these effects. Since real-world data is data collected as part of healthcare delivery, it inherently has these limitations because the scientific research questions it is being used to address were not framed at the time of data collection. Regardless, RWD plays a very critical role in addressing emerging research questions. Thus, during the COVID-19 pandemic, RWD played a crucial role in answering questions including  providing evidence to inform global public health decisions and improving clinical trials. The HealthVerity dataset used in this analysis is a unified database of open and closed medical and pharmacy claims, hospital transaction records for inpatient and outpatient hospital encounters, and laboratory data for individuals in the U.S. The amalgamation of data from multiple (individual) sources inherently results in a larger population size. However, Figure S1. in the supplementary section outlines the study flow diagram the authors used to identify individuals specifically meeting the inclusion and exclusion criteria.

Reviewer 3 Comment 2:   It has been repeatedly reported that the beneficial effects of the vaccine are age related.  The older you are, the greater the benefit.  Did they ever stratify their results on the basis of age?

Author Response: Specific stratification on age was not conducted, however, age adjustment in the propensity score model was conducted.

Reviewer 3 Comment 3:  Again, vaccine effectiveness is the key issue, and the authors have stated that the literature available to them is conflicting and limited in scope.  Their data can provide a general answer to the question of vaccine efficacy in a large proportion of the medical records from a substantial segment of the people of the United States.  Only with information as background can we properly interpret the significance of their findings regarding booster vaccine efficacy. 

Author Response: We thank the reviewer for this positive comment and agree with the points made.

  1. Response to additional comments from Reviewer 3:

Comment: In the discussion, the data above was compared to several limited studies of vaccine efficacy in general.  This comparison took up only one paragraph.  Interestingly, one of the studies reported no increased vaccine efficacy after vaccination

Author Response: The authors agree with this comment from the reviewer. There is a scarcity of studies that have specifically explored the effectiveness of Ad26.COV2.S when used as a homologous booster (2 doses of Ad26.COV2.S) or a heterologous booster (a single dose of Ad26.COV2.S plus an additional mRNA vaccine dose).

Comment: There was also a limited discussion of potential limitations.

Author Response: We thank the reviewer for this comment; however, the authors disagree with this assertion. The 4th, 5th, and 6th paragraphs of the "Discussion" section are explicitly dedicated to potential limitations of the study.

Comment: The issue of vaccine efficacy for COVID-19 remains a hot issue and is likely to remain so for the next generation.  This includes both medical professionals and the population in general, many of whom are still asking the question of what exactly happened in the vaccine response to COVID.  The authors themselves point out that the studies of vaccine efficacy that they identified in the literature were limited in scope and provided conflicting evidence of efficacy.  The authors’ use of HealthVerity information, which is tremendous in number, provides an exceptional opportunity for clarification of this issue, which is raised by the authors but not addressed from their own data.

Author Response: We thank the review for this comment and agree with the points made.

Comment: Again, vaccine effectiveness is the key issue, and the authors have stated that the literature available to them is conflicting and limited in scope.  Their data can provide a general answer to the question of vaccine efficacy in a large proportion of the medical records from a substantial segment of the people of the United States.  Only with information as background can we properly interpret the significance of their findings regarding booster vaccine efficacy.

Author Response: We thank the reviewer for this positive comment and agree with the points made.

Round 2

Reviewer 2 Report

Comments and Suggestions for Authors

The authors should write the explanation in the manuscript what they wrote in the  answers to reviewers concerns for point 2 for assessing after 152 days although it could have been after 180 days.  

Author Response

Reviewer Comment: The authors should write the explanation in the manuscript what they wrote in the  answers to reviewers concerns for point 2 for assessing after 152 days although it could have been after 180 days.  

Author Response: The following text has been added to the "results" section of the manuscript, "We used 152 days because at the time of protocol development and submission for IRB approval, the official FDA guidance of at least 6 months after the completion of a two-dose mRNA primary series for eligible individuals had not yet been developed. This (152 days) reflected the most scientifically accurate timeframe based on evolving data from clinical trials and scientific manuscripts in development ".